# Designing Ex Situ Conservation Strategies for *Butia capitata* [Mart. (Becc.) Arecaceae], a Threatened Palm Tree from Brazilian Savannah Biome, through Zygotic Embryo Cryopreservation

**Giuliano Carvalho Frugeri** [1], **Gabriela Ferreira Nogueira** [2], **André Luís Xavier de Souza** [3] **and Jonny Everson Scherwinski-Pereira** [3,*]

[1] Departamento de Botânica, Universidade de Brasília, Brasília 70919-970, DF, Brazil
[2] CNPq/Embrapa Postoctoral Fellowship Program, Embrapa Recursos Genéticos e Biotecnologia, Brasília 70919-970, DF, Brazil
[3] Embrapa Recursos Genéticos e Biotecnologia, Av. W5 Norte (Final), Brasília 70770-917, DF, Brazil
[*] Correspondence: jonny.pereira@embrapa.br

**Abstract:** Jelly palm (*Butia capitata*) is a species of palm tree endemic to the Brazilian Savannah (Cerrado) Biome, which stands out for its use as food, which has resulted in a predatory exploitation of its natural populations. This study aimed to characterize fruits and diaspores of natural populations of jelly palm ('Arinos', 'Mirabela' and 'Serranópolis'), in addition to developing ex situ conservation strategies of the species, through the storage of zygotic embryos at ultra-low temperatures of liquid nitrogen (LN). Initially, fruits, seeds and embryos were evaluated for their morphological characteristics. For conservation, hydrated zygotic embryos were desiccated for up to 12 h in a laminar-flow chamber and immersed directly in LN with different moisture contents. As a result, we observed morphological differences among the studied populations. The population 'Arinos' showed more expressive results for most of the evaluated characteristics, such as length, width of fruits and seeds. The cryopreservation of zygotic embryos proved to be efficient for the conservation of the species reaching germination rates between 70 and 86%, when the moisture content of the embryos immersed in LN were between 10 and 14%. The plants that reached the stage of the greenhouse had a rate of more than 80% survival. The variability in the characteristics of fruits and diaspores of the analyzed populations allow for establishing divergent groups for the definition of conservation strategies or genetic improvement. The developed cryopreservation protocol can be considered simple and efficient for the conservation of the genetic variability of the species.

**Keywords:** Arecaceae; biometric parameters; genetic resources; seed conservation

## 1. Introduction

Palm trees of the genus *Butia* are distributed throughout several South American countries [1], with several species present in the Brazilian territory [2]. The species *Butia capitata* stands out as being endemic to the Cerrado Biome, being found close to the border of the states of Goiás, Minas Gerais and Bahia [3]. Popularly known as jelly palm, this species is valued for its fruits, which can be consumed fresh or used in the preparation of juices, sweets, and ice cream. Additionally, its leaves have practical application in handicrafts [4].

However, the natural populations of *Butia* are currently facing several challenges including uncontrolled extractivism, deforestation, and intensive livestock. Additionally, the species faces mechanical barriers to germination due to the rigidity of the endocarp and dormancy, which hinders the regeneration of new plants [5–8] and limits multiplication for agronomic purposes. These factors collectively pose obstacles to conservation efforts

aimed at preserving the genetic variability of the species in field conditions. Magalhães et al. [9] warn of the genetic erosion of populations of *Butia capitata* in the Northern State of Minas Gerais, Brazil. In this regard, the implementation of germplasm management and conservation strategies for natural populations is emerging as a viable solution to protect and restore the species. These strategies not only mitigate the issues associated with excessive exploitation but also provide essential support for genetic improvement, cultivation, and propagation of the plants [10–13].

Seed conservation is regarded as one of the most effective ex situ approaches for preserving plant germplasm. However, the storage of seeds cannot be universally applied with equal efficacy to all plant species, as each species exhibits distinct characteristics concerning tolerance to reduced temperatures and low humidity, which are critical factors in conservation [14]. In a previously conducted study with *Butia capitata*, Dias et al. [15] found that the seeds were sensitive to storage at −18 °C for 90 days and, based on the results obtained, were classified as intermediate in terms of storage capacity, whereas Salomão and Santos [16] report that *B. capitata* seeds exhibit recalcitrant behavior. On the other hand, in studies conducted with *B. odorata*, Fior et al. [17] inferred that the seeds displayed orthodox behavior. They achieved a viability index above 90% with a water content of 6.14% and storage at sub-zero temperatures. These divergent results indicate that further studies need to be conducted to evaluate and define the best conservation strategies for the species.

The long-term storage of germplasm from species with recalcitrant and intermediate seeds is achievable through cryogenic technology [18]. Cryopreservation is a germplasm conservation method that involves maintaining plant material at ultralow temperatures, typically in liquid nitrogen (−196 °C), with or without cryoprotective substances [11–13]. At this temperature, cell divisions and metabolic processes are drastically reduced, leading to extended storage periods under ideal conditions. This prolonged storage capability, combined with reduced sample storage costs, makes cryopreservation an advantageous method for long-term germplasm conservation [19]. In cryopreservation, the risks of loss of biological material and genetic variability of the stored material are lower. In addition to these aspects, a cryobank requires minimal maintenance compared to other available systems, as cultures are stored in a compact space and protected from contamination [12,13]. Due to these factors, and especially due to the physiological characteristics of the majority of palm tree seeds, cryopreservation is the most used and recommended method for the conservation of germplasm from these species [20–23].

Considering the social, environmental and ecological importance of the species, coupled with genetic erosion, species endemism and scarcity of works, this study aims to characterize the fruits of different populations of *Butia capitata*, in addition to developing ex situ conservation strategies of the species, through the storage of zygotic embryos at cryogenic temperatures.

## 2. Materials and Methods

### 2.1. Plant Material

*Butia capitata* samples were collected during the summer months of November and January in the years 2013–2016. The diaspores were randomly selected for morphometric characterization, as well as for the cryopreservation experiments of zygotic embryos. In total, we studied three populations in the rural areas of the municipalities of Arinos (15°55′01″ S and 46°06′20″ W), Mirabela (16°16′2.31″ S and 44°11′45.14″ W), and Serranópolis (15°53′59″ S and 42°49′22.88″ W).

### 2.2. Fruit Characterization

To assess the biometric variability, fresh and mature fruits of each population were selected. Initially, we measured the longitudinal length and the equatorial diameter. A portion of the pulp was removed from the equatorial medial portion to allow for thickness measurements in three distinct regions. The entire pulp was then removed to measure

its mass and the humidity content. The fresh mass of the fruits was calculated by adding the mass of the fresh pulp and the fresh mass of the pyrenium. Measurements of the longitudinal length, width, and fresh mass of the pyrenium were taken before they were broken to examine endocarp thickness and endosperm separation. Subsequently, the humidity content of the endocarp and endosperm was determined separately. With the seeds still intact, we measured the longitudinal length and the equatorial diameter at the average portion. The embryo from the same seed was excised for the measurement of longitudinal length and equatorial diameter (mm).

Using the collected data, we calculated the proportion of pulp by dividing the mass of the pulp by the mass of the fresh fruit, and the proportion of the pyrenium, by dividing the mass of the pyrenium by the mass of the fresh fruit. The measurements of length and width were taken using a digital caliper. To determine the humidity content of the pulp, seed, and embryo, we employed an analytical precision scale and an oven with 105 °C air circulation for 24 h.

The experimental design employed was a completely randomized design with three replicates, consisting of ten individuals per replicate for each studied population. The collected data were subjected to analysis of variance, and mean comparisons were conducted using the Tukey test ($p > 0.05$) with the statistical program SISVAR [24]. Following analysis, a principal component analysis (PCA) was performed to determine whether the morphological descriptors were sufficient for group differentiation. Among the available descriptors, those exhibiting a coefficient of variation greater than 30% were selected, as they indicated high heterogeneity and were suitable for the PCA technique. Consequently, six descriptors were considered: fresh fruit mass, fresh pulp mass, dry pulp mass, pulp percentage, pyrenium percentage, and pyrenium mass.

### 2.3. Cryopreservation of Zygotic Embryos

For the cryopreservation experiment, seeds of *Butia capitata* from the three populations were disinfected by immersing them in 70% alcohol ($v/v$) for 3 min, followed by treatment with a sodium hypochlorite solution (2–2.5% active chlorine) for 30 min. The seeds were then rinsed three times with distilled and sterilized water. In a laminar-flow chamber, the zygotic embryos were carefully excised using tweezers, scalpels and dental pliers.

Initially, the zygotic embryos were hydrated in agar–water medium and kept in the dark at a temperature of 25 ± 2 °C for 16 h to standardize the humidity [23]. At this stage, the fresh mass (g) of the embryos was measured before and after hydration. Next, the zygotic embryos were exposed to airflow provided by the laminar-flow chamber for desiccation periods of 0, 2, 4, 6, 8, 10 and 12 h at approximately 25 °C. After each desiccation period, the embryos were either directly inoculated into a germination medium as a control treatment or placed in sterile cryotubes (2 mL) and immersed in liquid nitrogen (LN) for 48 h. Another sample of embryos was oven-dried at 105 °C for 24 h for humidity measurement, calculated as follows: $U\% = [(Pi - Pf)/Pi] \times 100$, where $Pi$ represents the initial sample mass (g), $Pf$ represents the final sample mass (g), and $U$ represents the water content in percentage on a wet basis (%).

The cryotubes were removed from the LN and quickly immersed in a water bath at 40 °C for 90 s. Subsequently, the embryos were aseptically transferred to test tubes containing 10 mL of germination medium. The germination medium consisted of salts and vitamins from the MS medium [25], supplemented with 30 g L$^{-1}$ sucrose and solidified with 2.5 g L$^{-1}$ Phytagel (Sigma, St. Louis, MO, USA). The pH was adjusted to 5.8 ± 0.1 before autoclaving at 121 °C and 1.5 atm pressure for 20 min.

The embryos were incubated in the dark for 21 days and then placed under a light intensity of 100 μm m$^{-2}$ s$^{-1}$ light, with a photoperiod of 16/8 h (light/dark) and a temperature of 25 ± 2 °C. After 60 days of culture, the germination of the zygotic embryos was evaluated. In this study, germination was defined as a biological process resulting in the resumption of growth leading to the emergence of the radicle. The development was considered complete when both the radicle and the aerial part were formed.

The experiment followed a 2 × 7 factorial design (cryopreservation × desiccation periods) resulting in a total of 14 treatments. Each population was composed of three replicates, and each replicate consisted of 10 zygotic embryos. The data were subjected to analysis of variance, and means comparisons were performed using Tukey test ($p > 0.05$) with the statistical program SISVAR [24].

### 2.4. Acclimatization

Plants exhibiting a healthy appearance, with roots measuring at least 2.0 cm in length and two or more leaves, were selected for the acclimatization process. At this stage, the plants were approximately 200 days old after germination.

To minimize plant loss, a pre-acclimatization procedure was carried out before transitioning the plants to greenhouse conditions. The plants were carefully removed from the culture medium and washed with running water to remove any excess media. Following cleaning, each plant was transplanted into transparent plastic cups with a capacity of 200 mL. The cups were filled with a mixture of sand and substrate (Bioplant, Nova Ponte, MG, Brazil) in a 1:1 ratio (*v/v*). Each cup was properly accommodated with another above (empty, to maintain humidity), which were fixed with tape at the edges. The cups were then placed in a germination chamber with a controlled temperature of 25 ± 2 °C and photoperiod of 16 h. The plants were manually irrigated every day for a duration of 30 days.

During the initial two-week phase, the plants were completely isolated by an additional plastic cup cover. In the final two weeks, the top of the cup (the additional cup over the plant) was removed, allowing the plant to be exposed to the surrounding environment while still being kept inside the germination chamber. After this period, the plants were transferred to a greenhouse to promote further development. In the greenhouse, the plants were transferred to 17 cm × 11 cm plastic bags filled with a substrate composed of red soil, washed sand, and bovine manure in a 3:1:1 ratio (*v/v*). Under these conditions, the plants were irrigated daily, ensuring that direct water jets did not hit the plants.

## 3. Results and Discussion

### 3.1. Fruit Characterization

In general, the results showed consistent developmental characteristics among the populations studied. The *Butia capitata* fruits exhibited a yellowish color, firm consistency, and smooth surface, which are typical of the mature stage. These fruits are oval-shaped drupes (Figure 1A), and some of them displayed an irregular equatorial diameter, likely due to multiple fruits developing in close proximity within a single cluster. The epicarp and the mesocarp present a fleshy and fibrous consistency, composing the pulp, which is the most commercially exploited part [26,27].

The rigid endocarp had a brown color and three longitudinal scars, with an operculum between each scar (Figure 1B). The seed had a brown surface with recesses and an appendage facing one of the opercula (Figure 1C). Regarding the seed, it displayed bilateral symmetry and a longitudinal scar in the area of the operculum (Figure 1D). The endosperm, which is white, had a central cavity forming an axis where the haustorium develops [28]. The zygotic embryo was located at one end of the axis, near the operculum (Figure 1E).

The collected fruits were dissected to measure the humidity content of the pulp, seed, and embryo. The fruits from the 'Serranópolis' population exhibited an average humidity content of 86.6% for the pulp, which was higher than the other populations. There was no statistical difference among the populations in terms of seed humidity content. However, for embryo humidity content, the populations of 'Arinos' and 'Serranópolis' differed significantly from 'Mirabela'; 'Arinos' presented the highest observed value (Table 1).

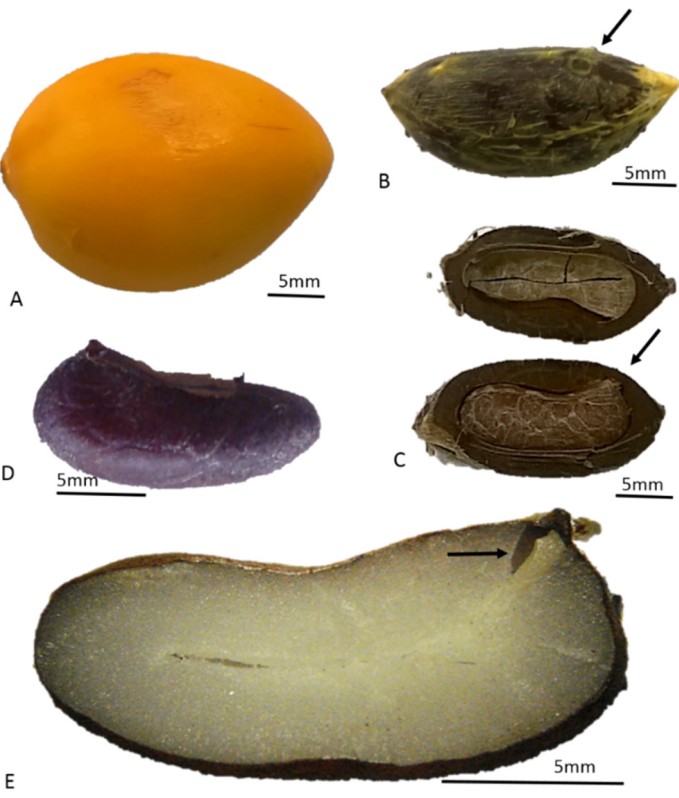

**Figure 1.** General appearance of fruit and diaspora of *Butia capitata*. (**A**): Fruit; (**B**): pyrenium (endocarp and seed), highlighting the operculum (arrow); (**C**): longitudinal cut of the endocarp, highlighting the position of the embryo; (**D**): seed; (**E**): longitudinal section of a seed, highlighting the embryo.

**Table 1.** Moisture content observed in pulp, seed and zygotic embryo of *Butia capitata* fruits collected in populations located in the north of Minas Gerais, Brazil.

| Population | Moisture Content * | | |
|---|---|---|---|
| | **Pulp** | **Seed** | **Zygotic Embryo** |
| Arinos | 79.0 ± 2.0 c | 6.2 ± 9.0 a | 18.4 ± 8.0 a |
| Mirabela | 82.4 ± 1.0 b | 5.3 ± 2.0 a | 7.5 ± 8.0 b |
| Serranópolis | 86.2 ± 2.0 a | 6.7 ± 8.0 a | 14.4 ± 8.0 a |
| CV (%) | 2.5 | 18.9 | 30.4 |

* Means ± SE followed by the same letter in the column do not differ by Tukey test ($p > 0.05$).

In general, significant morphometric differences were observed among the populations included in this study, with 'Arinos' showing the highest averages values for most of the analyzed variables (Table 2). The longitudinal length of the fruit differed for the three populations, with an average of 34.29 mm for 'Arinos', 28.37 mm for 'Serranópolis', and 26.25 mm for 'Mirabela', representing a difference of 23.5% between them. These values are similar to those reported by Silva and Scariot [8], who found average longitudinal lengths of 25.2 mm for 'Mirabela' and 25.4 mm for 'Serranópolis' in their studies conducted in 2006 and 2007. Pedron et al. [29], when analyzing the fresh fruit mass of *Butia odorata* populations in Santa Maria, RS, Brazil, reported variations from 5.6 to 26.4 g among them.

**Table 2.** Morphometric data of the fruit of three *Butia capitata* populations.

| Biometric Parameters | Populations | | | CV (%) | F |
|---|---|---|---|---|---|
| | Arinos | Mirabela | Serranópolis | | |
| Fruit length (mm) | 34.3 ± 0.2 A | 26.3 ± 0.1 C | 28.4 ± 0.2 B | 4.7 | 269.1 ** |
| Fruit width (mm) | 22.7 ± 0.2 A | 21.6 ± 0.2 B | 21.6 ± 0.2 B | 6.1 | 6.9 ** |
| Fruit fresh mass (g) | 8.2 ± 1.6 A | 6.5 ± 0.1 B | 6.6 ± 1.6 B | 14.9 | 24.78 ** |
| Pulp thickness (mm) | 4.8 ± 0.07 A | 4.7 ± 0.1 A | 4.8 ± 0.1 A | 12.2 | 0.3 ns |
| Pulp fresh mass (g) | 6.2 ± 0.1 A | 5.1 ± 0.1 B | 4.9 ± 0.1 B | 15.3 | 20.2 ** |
| Pulp dry mass (g) | 1.3 ± 0.03 A | 0.8 ± 0.02 B | 0.6 ± 0.03 C | 18.9 | 91.7 ** |
| Pyrenium length (mm) | 26.4 ± 0.2 A | 21.7 ± 0.2 C | 23.0 ± 0.2 B | 5.5 | 100.5 ** |
| Pyrenium width (mm) | 11.4 ± 0.2 A | 10.8 ± 0.1 B | 11.3 ± 0.1 AB | 7.8 | 4.5 * |
| Pyrenium fresh mass (g) | 2.0 ± 0.5 A | 1.3 ± 0.2 C | 1.6 ± 0.4 B | 21.5 | 24.5 ** |
| Endocarp thickness (mm) | 2.1 ± 0.07 B | 1.9 ± 0.06 B | 2.4 ± 0.05 A | 15.6 | 11.8 ** |
| Seed length (mm) | 16.0 ± 0.2 A | 12.4 ± 0.1 C | 13.8 ± 0.05 B | 7.7 | 83.2 ** |
| Seed width (mm) | 6.2 ± 0.1 A | 6.3 ± 0.1 A | 5.9 ± 0.1 A | 10.2 | 2.9 ns |
| Zygotic embryo length (mm) | 3.1 ± 0.07 B | 3.3 ± 0.1 AB | 3.5 ± 0.7 A | 13.6 | 4.5 * |
| Zygotic embryo width (mm) | 0.9 ± 0.02 B | 0.8 ± 0.03 B | 0.9 ± 0.03 A | 15.3 | 6.5 ** |

Means ± SE followed by the same letter in the row do not differ statistically from each other by the Tukey test at 5% probability level. * Significant ($p \leq 0.05$), ** highly significant ($p \leq 0.01$) and ns non-significant.

The equatorial diameter of the fruits was also higher for 'Arinos', with a mean of 22.74 mm. Previous studies conducted by Moura et al. [19] reported an equatorial diameter of 21.1 mm in populations from the municipality of Montes Claros, Brazil, while Silva and Scariot [8] reported values of 22.4 mm and 23.1 mm for populations of 'Mirabela' and 'Serranópolis'. In line with the findings of Moura et al. [27], who concluded that larger fruits are associated with greater pulp production, our data for the 'Arinos' population support this observation. The larger size of the fruits in this population corresponded to a greater fresh mass, both for whole fruit and for the pulp. However, the other populations, despite having lower values for these variables, still had averages similar to those described by Silva and Scariot [8]. In a study conducted by Rivas and Barilani [30] on fresh fruit mass of *Butia odorata* populations in Uruguay, a smaller variation was found, with averages ranging from 5.7 to 8.1 g.

The fresh mass of the pyrenium exhibited significant differences among populations. The 'Arinos' population had the highest average (2.01 g), followed by 'Serranópolis' (1.67 g) and 'Mirabela' (1.36 g), representing a difference of 17% and 33% compared to 'Arinos', respectively. Previous studies by Moura et al. [19] reported a mean fresh mass of the pyrenium as 1.62 g, while Silva and Scariot [8] found values of 1.1 g and 1.25 g for 'Mirabela' and 'Serranópolis', respectively.

In terms of endocarp thickness and embryo width, the 'Serranópolis' population exhibited the highest means. The percentages for 'Arinos' and 'Mirabela' were lower, with 12% and 17.8% for endocarp thickness, 10% and 5% for embryo length, and 11% and 12.5% for the embryo width, respectively. The variables of pulp thickness and seed width did not show significant differences between populations.

Barbosa et al. [31] observed considerable biometric variations in *Mauritia flexuosa* fruits, including size, shape, and color, within population. Silva and Scariot [8] concluded that biometric parameters varied among populations of *Butia capitata* in the Cerrado region of Brazil and that fruits could serve as criterion for seed selection in seedling production. Reis et al. [32] found that larger seeds of *Copernicia prunifera* exhibited faster protrusion of the cotyledonary petiole. Rivas and Barilani [30] utilized biometric analyses of *Butia* fruits to estimate the reproductive and productive capacity of palm trees in Uruguay.

Pulp exploitation is an important agronomic criterion with breeding implications. Nunes et al. [33] conducted a study comparing various genotypes of *Butia* from a population in Santa Vitória dos Palmares, RS, Brazil, and were able to identify individuals with superior pulp quality for fresh consumption. Pulp productivity can be utilized as an agronomic selection parameter to identify the best individuals of *B. capitata* for fruit extraction [8]. In this study, the fruits of the 'Arinos' population showed favorable characteristics for pulp utilization in industry, such as the length and the fresh fruit mass. Characterization studies

of this species are highly valuable, especially considering its endemism to the Cerrado and its classification within the *Butia* genus; however, scientific research on this subject is still limited [34]. Furthermore, these results are significant for the selection of desirable materials for the preservation of ex situ germplasm.

The principal component analysis (PCA) revealed that the two main components explain 97.69% of the total variation. Figure 2 presents a biplot illustrating these two main components. The 'Mirabela' group appears to be more restricted as having individuals that are quite homogeneous with each other, displaying a relatively high pulp percentage but smaller fresh fruit and pulp mass, as well as a lower pyrenium content compared to the individuals in the 'Arinos' and 'Serranópolis' groups. The 'Serranópolis' group exhibits similarities to the Arinos group, except for a lower dry pulp mass compared to the latter. Finally, concerning the 'Arinos' group, it demonstrates the highest dry pulp mass compared to the 'Mirabela' and 'Serranópolis' groups. If the economic objective is to increase dry pulp mass, then the 'Arinos' population may be recommended as the most suitable group in terms of this variable.

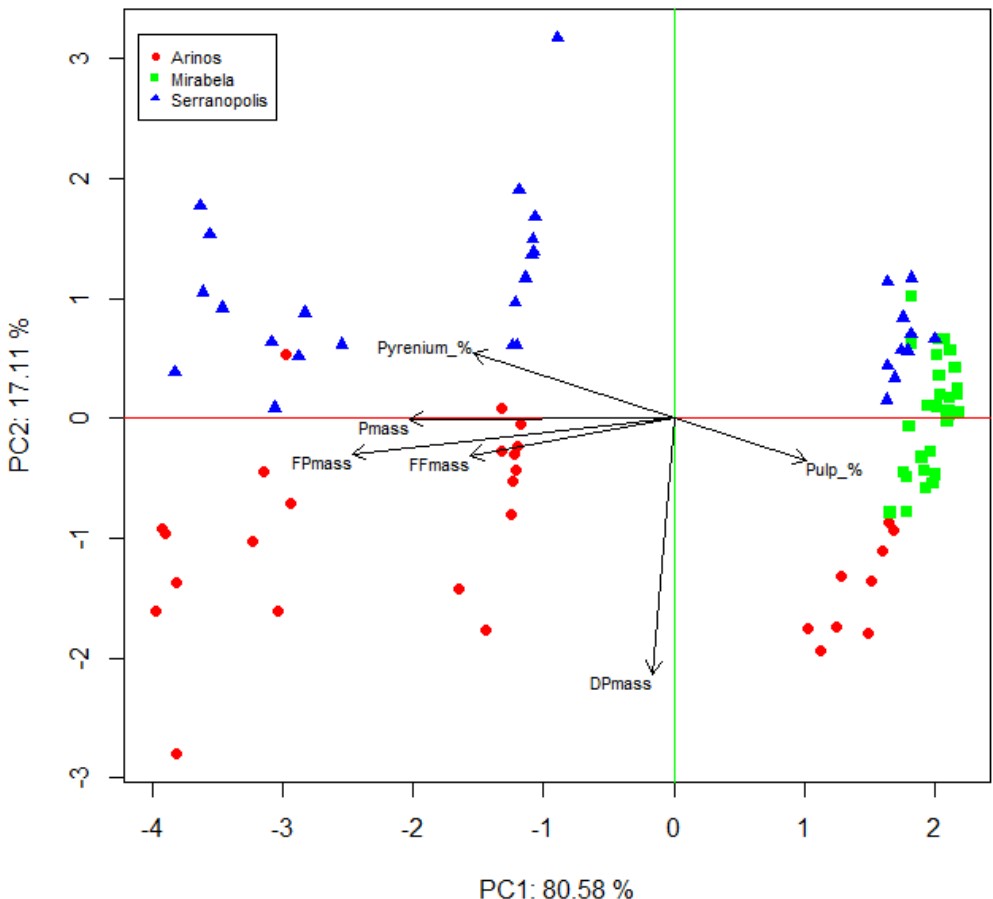

**Figure 2.** Biplot with the two main components. (Note: the length of the arrows representing the descriptors does not indicate their relative importance; they are solely for visualizing growth values).

*3.2. Cryopreservation of Zygotic Embryos*

After the hydration period, the initial moisture content of the zygotic embryos in the three populations was 69.6% for 'Arinos', 72.9% for 'Mirabela' and 77.8% for 'Serranópolis'. During the initial two hours of desiccation, in a laminar-flow chamber, there was a significant decrease in humidity across all populations, although it was gradual and stabilized at 8 h of flow exposure. Embryos from the 'Arinos' population displayed moisture content levels of 18.9%, 13.6%, 7.7%, and 6.9% between two and eight hours of airflow, stabilizing in subsequent time periods. 'Mirabela' embryos reached moisture contents of 25.9%,

14.2%, 10.4%, and 8.5% during the same period of time. Embryos from the 'Serranópolis' population exhibited moisture content levels of 26%, 18%, 9%, and 5.1% until stability was achieved (Figure 3).

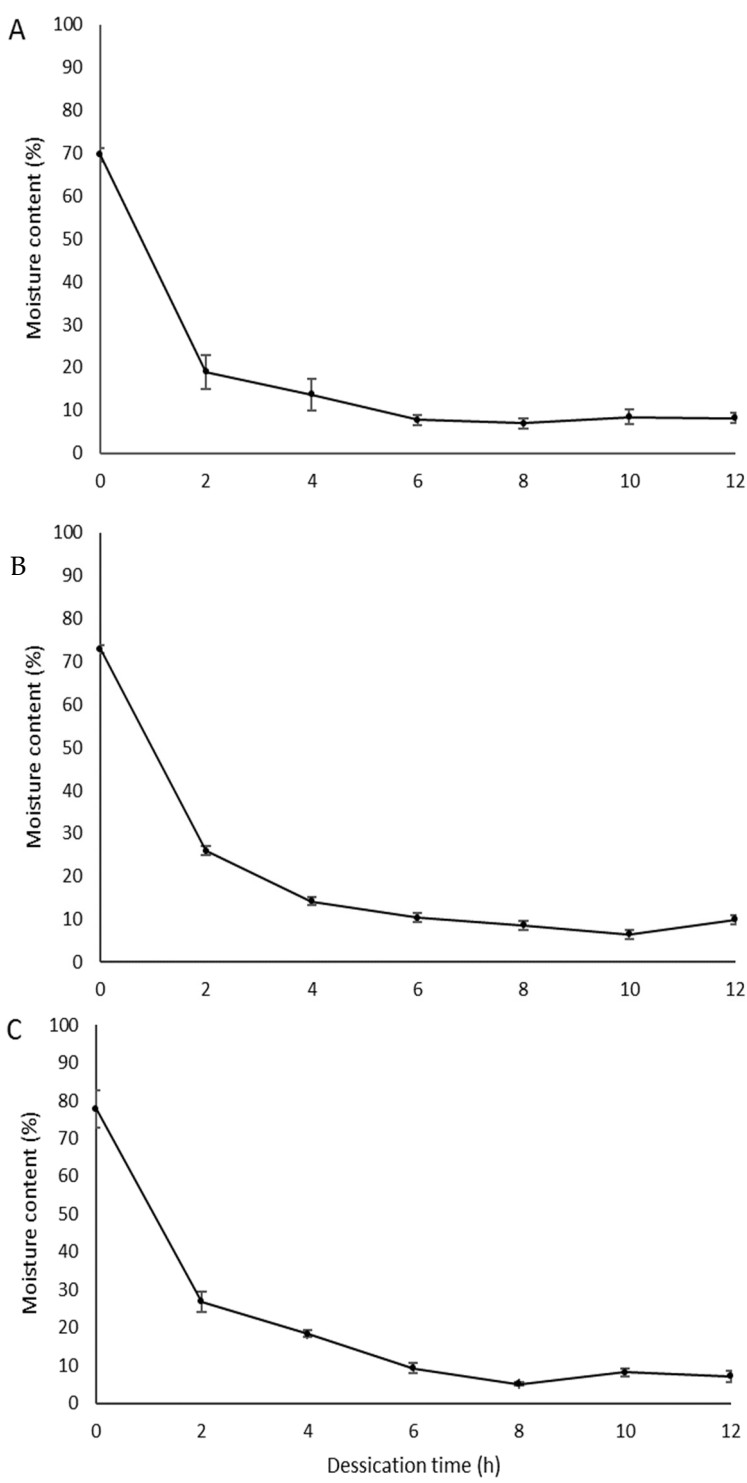

**Figure 3.** Moisture content of zygotic embryos of *Butia capitata* populations ((**A**): 'Arinos', (**B**): 'Mirabela' and (**C**): 'Serranópolis'), as affected by the desiccation time.

In this experiment, the control treatment, which involved embryos that were not cryopreserved, exhibited similar germination rates during the drying period in airflow.

However, for the embryos subjected to liquid nitrogen immersion, the germination rates were directly correlated with the moisture content of the embryos.

In the germination culture medium, the first observed morphological change in the excised zygotic embryos of *Butia capitata* seeds was swelling, characterized by an increase in size. The second morphological change was the elongation of the haustorium, accompanied by a curvature (Figure 4A). Following these two steps, radicle protrusion occurred, along with the emergence of the leaf sheath (Figure 4B). The achlorophyllous seedlings (Figure 4C) were transferred to a growth room and were exposed to light radiation, leading to the production of photosynthetic pigments (Figure 4D).

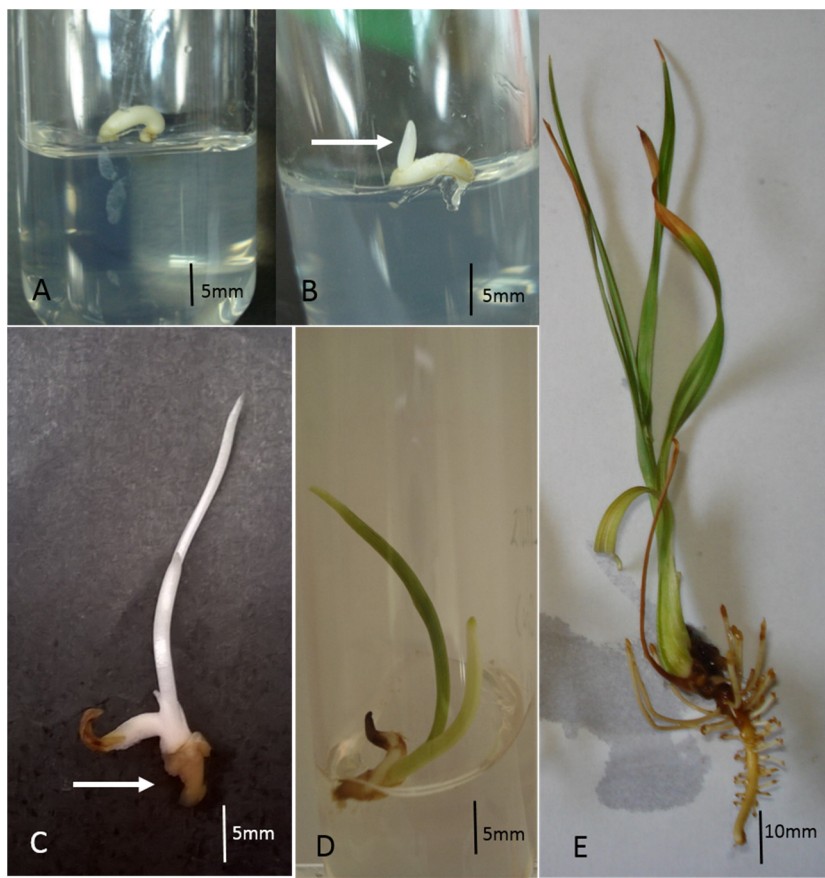

**Figure 4.** Stages of in vitro germination of *Butia capitata*. (**A**): Zygotic embryo 10 days after inoculation; (**B**): shoot protrusion; (**C**): seedling with developed shoot (arrow indicates radicle prominence); (**D**): pigmented shoot after exposure to light; (**E**): complete plant at 200 days after inoculation.

The initial germination rates observed for non-cryopreserved embryos were 76% for the 'Arinos' population, 76% for 'Mirabela' and 86% for 'Serranópolis'. However, for cryopreserved embryos at 0 h, there was no germination in any of the three populations due to the high moisture content of approximately 73%. After 2 h of desiccation and immersion in LN, the embryos from the 'Arinos' population showed a germination rate of 23%, which was statistically different from the embryos that were not subjected to nitrogen treatment (with an average germination rate of approximately 57%). From four hours of desiccation in airflow, the germination rates for cryopreserved embryos did not differ significantly from each other, maintaining a rate higher than 76%. These values were similar to those observed for non-cryopreserved embryos, except for the 12 h period, where the germination rate of the embryos exposed to the ultra-low temperature was higher than the control (-LN) (Table 3).

**Table 3.** Germination, shoot formation and complete plant from zygotic embryos of three *Butia capitata* populations submitted to different drying times and maintained (+LN) or not (−LN) in liquid nitrogen.

| Population | Desiccation Time (h) | Germination (%) * | | Aerial Part Formation (Shoot Only) (%) * | | Normal Plant Formation (Complete Plant) (%) * | |
|---|---|---|---|---|---|---|---|
| | | (−) LN | (+) LN | (−) LN | (+) LN | (−) LN | (+) LN |
| Arinos | 0 | 76.6 ± 8.0 aA | 0.0 bB | 46.6 ± 9.0 aA | 0.0 cB | 46.6 ± 9.0 aA | 0.0 cB |
| | 2 | 56.6 ± 9.0 aA | 23 ± 8.0 bB | 30.0 ± 8.0 aA | 16.6 ± 5.0 bcA | 30.0 ± 8.0 aA | 13.3 ± 4.0 bcA |
| | 4 | 70.0 ± 8.0 aA | 86.6 ± 6.0 aA | 50.0 ± 9.0 aA | 50.0 ± 9.0 abA | 40.0 ± 9.0 aA | 50.0 ± 9.0 aA |
| | 6 | 60.0 ± 9.0 aA | 76.6 ± 8.0 aA | 33.3 ± 8.0 aB | 60.0 ± 9.0 aA | 33.0 ± 8.0 aB | 60.0 ± 9.0 aA |
| | 8 | 63.3 ± 9.0 aA | 76.6 ± 8.0 aA | 26.6 ± 7.0 aA | 46.6 ± 9.0 abA | 26.0 ± 7.0 aA | 46.0 ± 9.0 abA |
| | 10 | 70.0 ± 8.0 aA | 80.0 ± 7.0 aA | 46.6 ± 9.0 aA | 60.0 ± 9.0 aA | 46.6 ± 9.0 aA | 60.0 ± 9.0 aA |
| | 12 | 60.0 ± 9.0 aB | 86.6 ± 6.0 aA | 30.0 ± 8.0 aA | 63.3 ± 9.0 aA | 26.0 ± 9.0 aA | 60.0 ± 9.0 aA |
| Mirabela | 0 | 76.6 ± 7.0 aA | 0.0 cB | 46.6 ± 9.0 aA | 0.0 ± 0 cB | 46.6 ± 9.0 aA | 0.0 ± 0 cB |
| | 2 | 56.6 ± 5.0 aA | 40.0 ± 5.0 bA | 30.0 ± 7.0 aA | 10.0 ± 3.0 cA | 30 ± 7.0 aA | 10 ± 3.0 cB |
| | 4 | 70.0 ± 5.0 aA | 43.3 ± 5.0 bB | 50.0 ± 8.0 aA | 26.6 ± 7.0 cbB | 43.3 ± 9.0 aA | 23.3 ± 6.0 bcA |
| | 6 | 60.0 ± 5.0 aB | 93.3 ± 3.0 aA | 33.3 ± 8.0 aB | 66.6 ± 8.0 aA | 30 ± 8.0 aB | 63.3 ± 8.0 aA |
| | 8 | 60.0 ± 5.0 aA | 73.3 ± 4.0 abA | 26.6 ± 7.0 aB | 53.3 ± 9.0 abA | 26.6 ± 7.0 aB | 53.3 ± 9.0 abA |
| | 10 | 70.0 ± 5.0 aA | 66.6 ± 5.0 abA | 46.6 ± 9.0 aA | 46.6 ± 9.0 abA | 43.3 ± 9.0 aA | 46.6 ± 9.0 abA |
| | 12 | 60.0 ± 5.0 aA | 60.0 ± 5.0 abA | 30.0 ± 7.0 aA | 26.6 ± 7.0 cbA | 26.6 ± 7.0 aA | 23.3 ± 7.0 bcA |
| Serranópolis | 0 | 86.6 ± 6.0 aA | 0.0 cB | 73.3 ± 7.0 aA | 0.0 ± 0 cB | 70 ± 7.0 aA | 0.0 ± 0 cB |
| | 2 | 70.0 ± 8.0 aA | 36.5 ± 9.0 bB | 56.6 ± 9.0 aA | 16.6 ± 5.0 cbB | 56.6 ± 9.0 aA | 16.6 ± 5.0 bcB |
| | 4 | 70.0 ± 8.0 aA | 76.6 ± 8.0 aA | 56.6 ± 9.0 aA | 56.6 ± 8.0 aA | 53.3 ± 9.0 aA | 56.6 ± 8.0 aA |
| | 6 | 83.3 ± 7.0 aA | 60.0 a ± 9.0 bB | 60.0 ± 9.0 aA | 50 ± 9.0 abA | 56.6 ± 9.0 aA | 43.3 ± 8.0 abA |
| | 8 | 86.6 ± 6.0 aA | 73.3 ± 8.0 aA | 76.6 ± 7.0 aA | 66.6 ± 8.0 aA | 73.3 ± 7.0 aA | 63.3 ± 8.0 aA |
| | 10 | 63.3 ± 9.0 aA | 70.0 ± 8.0 aA | 50.0 ± 9.0 aA | 63.3 ± 8.0 aA | 46.6 ± 8.0 aA | 63.3 ± 8.0 aA |
| | 12 | 73.3 ± 8.0 aA | 60.0 a ± 9.0 bA | 66.6 ± 8.0 aA | 66.6 ± 9.0 aA | 66.6 ± 9.0 aA | 50 ± 9.0 abA |

Means ± SE followed by the same uppercase letter in the row and lowercase letter in the column, within the same variable and population, do not differ statistically from each other by the Tukey test at 5% probability level.

Embryos excised from the 'Mirabela' population and immersed in LN for between 2 and 4 h exhibited the lowest germination averages, with rates of 40% and 43%, respectively, which were not statistically different from each other. The germination rate increased in the 6 h period, where the embryos with approximately 10% moisture reached a radicle emission rate of 93%. In desiccation periods longer than 6 h, the germination averages remained stable in relation to cryopreserved or non-cryopreserved embryos (Table 3). For the 'Serranópolis' population, embryos excised and immersed in LN showed an average germination rate of 36% after 2 h of desiccation in airflow, which was lower compared to other time intervals. After 4 h, the germination rate increased to 76% and remained relatively consistent in the following periods.

The results of cryopreservation in *Butia capitata* demonstrated that the best germination rates were achieved after 4 h of zygotic embryo desiccation, when the moisture content was below 18% in all three populations. This suggests that the species exhibits tolerance to desiccation at low levels (down to 5–6%) and storage at ultra-low temperatures. While Dias et al. [15] suggest the use of cryoprotectants to enhance cryopreservation efficiency in excised embryos, this study observed high germination rates similar to the control treatment, indicating that cryoprotectants may not be necessary.

Other palm species also show desiccation tolerance and successful storage in LN (−196 °C). N'Nan et al. [35] concluded that cryopreservation of *Cocos nucifera* embryos after silica gel drying is feasible and facilitates phytosanitary control for the establishment of germplasm collections. Similarly, Sisunandar [36] observed that embryo maturity is related to germination rates and complete plant formation in *Cocos nucifera* embryos subjected to LN. Cryopreservation of seeds with *Elaeis guineensis* integument proved effective, suggesting that rescuing zygotic embryos or using cryoprotectants may not be necessary [37]. Excised embryos of *B. gasipaes* dehydrated for 4 h, encapsulated, and rapidly immersed in LN retained viability [38]. According to Dickie et al. [39], the behavior of palm seeds in storage at ultra-low temperatures, associated with humidity content, may be influenced

by local factors in their natural habits. These authors suggest that orthodox species usually belong to dry habitats, where their seeds are naturally exposed to low humidity and that recalcitrant and intermediate species are found in environments where there are no significant humidity variations.

The development of completed plants is associated with the initial emergence of the leaf sheath and followed by eophil formation (Figure 4). Germination rates were always higher than the percentages of shoot development and complete plant formation intervals and for all populations, indicating that the formation of complete plants requires eophil development (Table 3). Martins-Corder and Saldanha [40] observed that the success of seedling survival in *Euterpe edulis* species is correlated with optimal shoot development. These results demonstrate the viability of cryopreservation techniques for ex situ germplasm maintenance of various palm species. This study established an ideal range of moisture for LN exposure of zygotic embryos, resulting in high germination rates, and provides a simple protocol that can be applied for species conservation.

*3.3. Acclimatization*

During the pre-acclimatization stage, survival rates of 82% were observed in plants from the cryopreservation experiments of zygotic embryos, within the initial days in the growth chamber. Gradual exposure to external conditions, known as pre-acclimatization, is one of the strategies to increase success during the acclimatization phase. After 40 days in the growth chamber, the surviving plants were transferred to a greenhouse for further development and completion of the acclimatization process. In general, the plants were tolerant to the new cultivation conditions in this second phase, with a survival rate greater than 90%.

Plants grown in vitro develop in a culture medium that differs in physical and chemical aspects from natural soil. In studies conducted by Ledo et al. [41] with *Cocos nucifera*, in vitro germinated embryos were transferred to a substrate containing sand after 210 days, followed by acclimatization in a greenhouse. Angelo et al. [42], working with *Elaeis guineensis* and *E. oleifera*, obtained better acclimatization results for plants with a well-developed root system from in vitro germinated zygotic embryos of both species. Padua et al. [43] verified that *E. guineensis* plants exhibited a higher acclimatization rate when they had well-developed roots at the moment of transfer from the culture medium to the substrate.

In addition to plant-related factors, acclimatization success is also related to the container, shading, substrate, leaves and root length [44,45]. In this work we observed factors such as the selection of plants with roots above 2 cm in length and shoots with two or more leaves and a pre-acclimatization stage in germination chambers may have directly contributed to the survival rates during this complex transition phase from the in vitro medium to the ex vitro condition.

**4. Conclusions**

In summary, we found that *Butia capitata* fruits collected from the populations of 'Arinos', 'Mirabela' and 'Serranópolis' show variation in their morphometric values for the observed characteristics. The fruits from 'Arinos' showed, in general, superior values when compared with the populations of 'Mirabela' and 'Serranópolis'. This result is relevant for being a species which is little studied, endemic, and threatened by predatory and commercially important extractivism, mainly in local populations. In addition, it provides subsidies for the maintenance of the germplasm of elite materials in the medium to long term. In addition, we presented a simple and efficient conservation protocol of the species since the zygotic embryos of *B. capitata* are tolerant to the ultra-low temperature of LN, and are able to be cryopreserved, provided they are within an optimal humidity range, preferably between 10 and 14%.

**Author Contributions:** J.E.S.-P. conceived and designed the experiments. G.C.F., G.F.N. and A.L.X.d.S. performed the experiments. G.C.F., G.F.N. and J.E.S.-P. analyzed the data and wrote the manuscript. All authors have read and agreed to the published version of the manuscript.

**Funding:** This research was funded by the Conselho Nacional de Desenvolvimento Científico e Tecnológico (CNPq Grant 426637/2016-0) and Coordenação de Aperfeiçoamento de Pessoal de Nível Superior (Capes/Embrapa 001-2011/Grant 39).

**Institutional Review Board Statement:** Not applicable.

**Informed Consent Statement:** Not applicable.

**Data Availability Statement:** Not applicable.

**Acknowledgments:** We thank Aldicir O Scariot for assistance in collecting biological material, Zanderluce G. Luis for technical assistance during the experiment, and Luis Alberto Martins P. de Melo for statistical help.

**Conflicts of Interest:** The authors declare no conflict of interest.

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
