# Peer review of "Designing Ex Situ Conservation Strategies for Butia capitata [Mart. (Becc.) Arecaceae], a Threatened Palm Tree from Brazilian Savannah Biome, through Zygotic Embryo Cryopreservation"

_2037-0164, doi:10.3390/ijpb14030047_

Round 1

Reviewer 1 Report

This MS is interesting in that it studies the morphological differences of three populations of Butia capitata and the cryopreservation of their zygotic embryos.

To improve the quality and interest of the MS, it would be interesting to present a brief summary of the literature on cryopreservation of embryos of palm species.

It would be also interesting to raise hypotheses to explain the differences in morphology and tolerance to cryopreservation of the three populations studied.

Detail comments:

17-18 : ultra-low temperature

43-44 : sentence impossible to understand

57 : arresting metabolism

356 : down to 5-6%

The MS should be edited by a professional translator/editor to improve the quality of English.

Reviewer 2 Report

Dear authors,

I have revised your manuscript “Designing ex-situ conservation strategies for Butia capitata [Mart. (Becc.) Arecaceae], a threatened palm tree from Brazilian Savannah Biome, through zygotic embryo cryopreservation" submitted for publication in International Journal of Plant Biology. This is, in general, an interesting paper on the development of a simple and rapid cryopreservation method for the long-term storage of Butia capitata embryos. I have suggested some edits below that could further improve the quality of the manuscript prior to publication. The paper is overall well-written but requires moderate revision to fix some blank spots (particularly in the introduction and methodology) and typos – see my remarks below.

Lines 17-18: at subzero temperatures of liquid nitrogen (LN)

Line 20: in LN

Line 20: Consider deleting “with different moisture contents”

Line 25: in LN

Line 27: allow for establishing

Lines 43-44: , it hinders the germination and regeneration of new plants [6-8]

Line 48: Add more references on cryopreservation. There are a lot of new and great manuscripts on this topic. Please find below some suggestions:

https://doi.org/10.1007/s11240-020-01770-0

https://doi.org/10.1007/s11240-020-01846-x

https://doi.org/10.3390/plants10122744

Line 59: Add a paragraph about explants used for cryopreservation. When to use seeds, embryos, or shoot tips? Add a paragraph on cryopreservation techniques

Line 67: Add the season

Line 105: How big were the embryos? (Add images)

Line 112: How about room temperature during dissection?

Line 144: How many embryos per cryotube?

Line 143: Was the substrate sterilized?

Line 154: Consider adding images about protocol steps…(embryo excision, dehydration, cryopreservation, regrowth, acclimatization)

Line 155: Results and Discussion

Lines 190 and 193: Check the italics in the scientific name throughout the manuscript

Table 1: Is the moisture content in percent?

Line 258: “fell”… poor expression. Why not use reduced?

Lines 262-264: Rewrite the sentence adding the values in their respective dehydration times

Line 334: at 0 hours of xxxxx

Line 336: immersion in LN

Line 342: Add the values

Lines 343 and 349: in LN

Line 357: “cryoprotectants”… this is the first time you mentioned it… please add a few examples and explain the main difference between the protocol you are proposing to this one..

Lines 360: in LN

Lines 361-363: Rewrite this sentence and explain the main difference between the protocol you are proposing to this one..

Lines 365 and 368: LN

Line 368: encapsulated in xxx

Line 383: in LN

Lines 400 and 402: et al was italicized in some places and not in others. ... check it through the manuscript

Line 408: container?

Line 423: in LN

What was the difficulty in cryopreserving this species? Consider adding this information.

What cryopreservation technique was used

What is the main advantage of cryopreservation to conserve this species?

Describe the best dehydration procedure to cryopreserve this species

I expected the Mirabella material to require less dehydration time, as the moisture content of the Zygotic embryo was lower than that of the other materials... Please add a comment on this point.

Please review the references following journal guidelines.

Round 2

Reviewer 2 Report

I can see an improvement in the quality of the manuscript. The authors attended to all comments and suggestions.